

**Real-time monitoring of nitrate transport in deep vadose zone under a crop**
**field—implications for groundwater protection**
T. Turkeltaub[a]*, D. Kurtzman[b], O. Dahan[a]
[a] Department of Hydrology & Microbiology, Zuckerberg Institute for Water Research,
Blaustein Institutes for Desert Research, Ben Gurion University of the Negev, Sde
Boker Campus, Negev 84990, Israel
[b] Institute of Soil, Water and Environmental Sciences, The Volcani Center,
Agricultural Research Organization, P.O. Box 6, Bet Dagan 50250, Israel
*Corresponding author (tuviat@post.bgu.ac.il)
Tel.: 972-8-6485012; Mobile: 972-52-3584844; Fax: 972-8-6563504



**Abstract**
Nitrate is considered the most common non-point pollutant in groundwater. It is often
attributed to agricultural management, when excess application of nitrogen fertilizer
leaches below the root zone and is eventually transported as nitrate through the
unsaturated zone to the water table. A lag time of years to decades between processes
occurring in the root zone and their final imprint on groundwater quality prevents
proper decision-making on land use and groundwater-resource management. In this
study, water flow and solute transport through the deep vadose zone underlying an
agricultural field were monitored using a vadose-zone monitoring system (VMS).
Data obtained by the VMS over a period of 6 years allowed detailed tracking of water
percolation and nitrate migration from the surface through the entire deep vadose zone
to the water table at 18 m depth. The temporal variations in the vadose zone sediment
water content were used to evaluate the link between rain patterns and water fluxes. A
nitrate concentration time series, which varied with time and depth, revealed—in real
time—a major pulse of nitrate mass propagating down through the vadose zone from
the root zone toward the water table. Analysis of stable nitrate isotopes indicated that
manure is the prevalent source of nitrate in the deep vadose zone, and these isotopes
were barely affected by natural soil or industrial nitrogen components. Total nitrate
mass estimations and simulated pore-water velocity using the analytical solution of
the convection–dispersion equation indicated dominance of nitrate vertical transport,
and excluded the possibility of lateral nitrate input. Accordingly, prevention of
groundwater pollution from surface sources such as agriculture has to include
effective and continuous monitoring of the entire vadose zone.






*Keywords:* Nitrate transport, Deep percolation, Vadose zone, Groundwater pollution





## 1 Introduction



Groundwater contamination by nitrate originating from agricultural land use is
a global problem. The World Health Organization guideline for maximum level of
nitrate in the drinking water is 50 mg/L $NO_3$. The US Environmental Protection
Agency (EPA) regards nitrate as requiring immediate action whenever its
concentration exceeds drinking-water standards (US EPA, 1994). A detailed
framework was established by the Nitrate Directive of the EC (European Community,
1991) to prevent water pollution by nitrate. Nevertheless, nitrate contamination has
disqualified drinking-water wells in Israel (local standard: 70 mg/L $NO_3$) more than
any other contaminant at the beginning of the 21st century (Elhanany, 2009). To
prevent excessive leaching of nitrate and its arrival to the groundwater, it is essential
to investigate and quantify the mechanism controlling nitrate migration in the
unsaturated zone with respect to the specific agrotechnical regime implemented on
land surface.
Nitrate fate in the subsurface has been investigated by various approaches,
such as: (i) isotopic signature in groundwater systems (Kaplan and Magaritz, 1986;
Wassenaar, 1995; Oren et al, 2004; Wassenaar et al., 2006; Showers et al., 2008;
Dejwakh et al., 2012; Baram et al., 2013), (ii) crop-management strategies, which
combine crop production and nitrate leaching to the subsurface (Leenhardt et al.,
1998a, 1998b; Hanson et al., 2006; Doltra and Muñoz, 2010; Beggs et al., 2011), and
(iii) studies based on data from the deep vadose zone (Onsoy et al., 2005; Green et al.,
2008; Dann et al., 2010; Nolan et al., 2010; Botros et al., 2012; Kurtzman et al., 2013;
Dahan et al., 2014; Turkeltaub et al., 2014, 2015a, 2015b).





Nitrogen is an essential nutrient for crop growth and is widely used as a
fertilizer. Although the dominant forms of soluble nitrogen fertilizer are reduced (e.g.
urea, ammonia) and evolve into other nitrogen species through biochemical processes
in the soil, nitrate is the most common chemical contaminant in the deep vadose zone
and groundwater (Green et al., 2008; Rupert, 2008).The transfer time of nitrate within
the deep vadose zone has been estimated to take from weeks to decades, depending on
the water regime, thickness of the unsaturated zone and lithological characteristics of
the subsurface (Spalding et al., 2001; Scanlon et al., 2010). Moreover, estimates of
cumulative nitrate fluxes in the unsaturated zone have shown significant differences in
the timing and magnitude of fluxes derived from different land uses (Green et al.,
2008; Dahan et al., 2014; Turkeltaub et al., 2014, 2015a, 2015b). Therefore, the
cumulative impact of nitrate leaching from the root zone through the unsaturated zone
on nitrate level in the groundwater is blurred by mixing and dilution in the aquifer
water. The tendency toward elevated nitrate concentration in aquifer water is thus a
relatively slow process (Green et al., 2008).
Although sampling groundwater from wells is easy, the concentration of
nitrate might already be at levels that will lead to disqualification of the aquifer water.
Knowing the time lag between initiation of a pollution process in the unsaturated zone
and its final impact on aquifer quality could give decision-makers more time to plan
possible backups for alternative water supply (Baram et al., 2014). Accordingly,
estimations of water and solute fluxes which are based on data from the deep
unsaturated zone could better indicate their potential long-term impact on the
groundwater (Scanlon et al., 2002; Onsoy et al., 2005; Green et al., 2008).
Nevertheless, most of these estimates are based on data obtained from excavated soil
profiles or pore-water sampling over a short period of time, which represent a





snapshot in time of the sediment's chemical state rather than dynamic temporal
variations. Furthermore, knowledge of nitrate's fate and transport below the root zone
is restricted due to issues such as soil spatial variability and long travel times in the
deep vadose zone (Onsoy et al., 2005).

The recent development of a vadose-zone monitoring system (VMS) enables

continuous monitoring of the hydrological and chemical properties of percolating
water in the deep vadose zone. Data collected by the system comprise direct
measurements of the water-percolation fluxes and the chemical evolution of the
percolating water across the entire unsaturated domain. To date, the VMS has been
successfully implemented in numerous studies on water flow and contaminant
transport in the unsaturated zone in a variety of hydrological setups, including: (i)
floodwater percolation in arid environments (Dahan et al., 2007, 2008, 2009), (ii)
rainwater percolation through thick sand and clay formations (Rimon et al., 2007;
Baram et al., 2012; Turkeltaub et al., 2015a), (iii) solute transport in the vadose zone
(Rimon et al., 2011; Baram et al., 2013; Dahan et al., 2014; Turkeltaub et al., 2014,
2015b), and (iv) impact of agriculture on groundwater quality (Turkeltaub et al.,
2014, 2015b).
A VMS was installed under a commercial crop field to study water flow and
nitrate transport through the deep vadose zone with respect to rain pattern as well as
the agricultural and fertilization setup. Continuous data on variations in the sediment
water content and nitrate concentrations were collected from the entire vadose zone
(18 m deep) for over 6 years. The temporal variations in the vadose zone sediment
water content were used to evaluate the link between rain pattern and water fluxes
(Turkeltaub et al., 2014). The nitrate concentration time series, which included
variation of nitrate in time and at multiple depths, revealed, in real time, a major pulse





of nitrate mass propagating down through the vadose zone toward the water table.
Stable nitrate isotope analysis identified the source of the nitrate in the subsurface.
The vertical transport properties and total nitrate mass in the vadose zone were
estimated according to the transient data. Results from the long-term monitoring were
compared with earlier modeling efforts applied at the same site (Turkeltaub et al.,

2014).


**2 Methods**

**2.1 Study area**

The study was conducted under a commercial crop field located in the
southern part of the coastal plain of Israel and situated on the outcrops of a phreatic
aquifer (34°41'13" E; 31°49'42" N). Mediterranean climate prevails in this area, with
hot, dry summers (May–September) and rainy winters (October–April), an average
annual rainfall of 512 mm and average temperatures of 31.2 °C (August) and 17.8 °C
(January) in the hottest and coldest month, respectively (Israeli Meteorological
Service, 2015). Reference evapotranspiration rates calculated according to the
Penman–Monteith method (suggested by the Food and Agriculture Organization)
range from 1.5 mm/day (January) to 5.7 mm/day (July) (Israeli Meteorological
Service, 2015).
From 2009 to 2013, the crop field site was cultivated with rainfed winter
crops—spring wheat (*Triticum aestivum* L.) and pea (*Pisum sativum* L.) (Fig. 1).
Then, for 1 year (2013/2014), the field was fallow. The crops were sown at the
beginning of the wet season (November) and grew into the spring (April) with no





additional irrigation. After harvest, plowing practice was implemented. Main
fertilization application to the crop field was dairy-farm slurry manure. In September
2014, due to a change in the agricultural cultivation type, jojoba (*Simmondsia*
*chinensis*) shrubs were planted, irrigation systems were installed, and the distribution
of manure ceased.

**2.2  Monitoring setup**

The crop field site was selected as representative of the prevalent agricultural
setting on the aquifer outcrops and was instrumented with a VMS (Fig. 1). Full
technical descriptions of the VMS structure, performance and installation procedures
can be found in other publications (Rimon et al., 2007, 2011; Dahan et al., 2008,
2009). For brevity, only a general description is given here.
The VMS is composed of a flexible sleeve installed in uncased slanted ($35^{\circ}$)
boreholes hosting multiple monitoring units at various depths. Each monitoring unit
has a flexible time-domain reflectometry (FTDR) sensor for continuous measurements
of sediment water content, and vadose-zone sampling ports (VSPs) for frequent
collection of pore-water samples from the unsaturated zone (Table 1). The slanted
installation ensures that each monitoring unit faces an undisturbed sediment column
that extends from land surface to the probe or sampling port depth. After insertion of
the VMS into the borehole, the flexible sleeve is filled with a high-density solidifying
material—liquid two-component urethane—that solidifies in the borehole shortly after
its application, thereby ensuring proper sleeve expansion for attachment of the
monitoring units to the borehole's irregular walls, sealing its entire void and
preventing potential cross-contamination by preferential flow along the borehole.





The VMS under the crop field included eight monitoring units distributed

vertically and laterally along the entire vadose zone cross section from a depth of 1 m
to a depth of 18 m. Note that each monitoring unit is shifted vertically and
horizontally from the others along the slanted orientation of the installation (Fig. 1).
Since each monitoring unit is located under its own undisturbed sediment column, the
integrated data from the VMS should be regarded as representative of a wider zone
rather than a single vertical profile. Sediment water content was monitored daily. The
pore-water sampling campaigns were conducted every 90 days on average.

**2.3 Nitrate-transport simulations**

The observed nitrate concentration dynamics at the 6.3 m depth (Table 1) was

analyzed and compared with earlier modeling estimations conducted according to
observations of water content under the crop field (Turkeltaub et al., 2014). Nitrate
transport was modeled in terms of the convection–dispersion equation (CDE)
equilibrium assuming resident concentration for a third-type inlet condition as follows
(Toride et al., 1999):
$$R\frac{\partial c}{\partial t} = D\frac{\partial^2 c}{\partial x^2} - v\frac{\partial c}{\partial x} \qquad (1)$$

where $c$ is the solution concentration, $x$ is distance, $t$ is time, $D$ is the dispersion
coefficient, $v$ is the average pore water velocity (water flux $q$ divided by the water
content $\theta$), and $R$ is the retardation factor.
The nitrate concentrations obtained by the VSP located at the 4.2 m depth (Table 1)
served as a series of successive applications of solute pulses (multi-pulse boundary
condition). Both VSPs were located in a relatively homogeneous medium of sandy
texture (Turkeltaub et al., 2014), following the intrinsic assumption of CDE analytical





model homogeneity. The CXTFIT2 code (Toride et al., 1999) and the Levenberg–
Marquardt-type optimization approach (Marquardt, 1963), both included in
STANMOD (van Genuchten et al., 2012), were used for inversely estimating the
pore-water velocity ($v$) and dispersion coefficient ($D$) according to observed
concentrations. Both parameters were obtained by running CXTFIT2 multiple times
for inverse optimization, each time with different initial values (Turkeltaub et al.,
2015a,b).

**2.4  Total nitrate mass**

The total nitrate mass in the unsaturated zone estimations was calculated to

emphasize the nitrate mass that will eventually contaminate the groundwater. The
following equation was used for yearly nitrate mass (per area) in the vadose zone:
$$M = \int_{Z=water\_table}^{Z=ground\_surface} \overline{\theta_i} \times C_i \times dz_i \tag{2}$$

where $M$ is nitrate mass in the vadose zone under a unit area, $i$ indexes the depth
interval for which the corresponding VSP is at its centre, $C_i$ is the nitrate
concentration [$M/L^3$] sampled with the VSP at that depth interval, $\theta i$ is the average
water content measured by the nearest FTDR sensor, and $dz_i$ is the interval length
(Fig. 2).

**3  Results and discussion**

**3.1  Nitrate migration in the unsaturated zone**





From September 2009 to the time of the study, water contents were monitored

at multiple depths by the VMS in the vadose zone of the crop field. The continuous
monitoring of the vadose zone indicated temporal variations in measured water
contents (Fig. 2). Throughout the monitoring period, most of the rainstorms caused a
rise in the water content measured by the shallowest water sensor (0.5 m, Fig. 2). At
the 2.1 m and 3.1 m depths, the rise in water contents corresponded mainly to
significant rain events (Fig. 2b,c). The sensors at the deeper depths displayed temporal
variability with respect to the cumulative annual rain pattern. In some years, a lag
between the end of the rainy season and the rise in water content was recorded,
whereas in other years, the rise in water content occurred throughout the entire vadose
zone following a significant rain event (Fig. 2d–h). A more detailed description of the
sequential rise in water content with depth following a wetting event on land surface,
and a clear indication of propagation of a wetting wave down through the vadose zone
are presented in our earlier study at the site (Turkeltaub et al., 2014), and in other
studies at different sites as well (Rimon et al., 2007, 2011; Dahan et al., 2008, 2009;
Baram et al., 2012, 2013).

During the monitoring period, slurry from a nearby dairy farm was often

spread over the field, serving as a nitrogen source (Fig. 1). This practice was stopped
in 2014 when the agricultural setting in the field was changed to a jojoba orchard. The
dairy slurry was distributed throughout May and June, 60 days after harvesting, and
was scattered across the crop field.

A time series of nitrate concentrations at various depths under the field was

obtained by frequent sampling of the vadose zone pore water using the VSPs at
multiple depths (Fig. 3). Throughout 6 years of continuous monitoring, different
scales and magnitudes of the variations in nitrate concentration were observed. An





overview of the nitrate concentration time series with depth (Fig. 3) reveals a major
pulse of elevated concentrations, initiating close to the surface, and gradually
progressing down the vadose zone toward the water table at a depth of ~18 m.  The
process was first monitored at the uppermost VSP at 1 m depth, where nitrate
concentrations displayed a significant increase during the winter of 2010/2011. Then a
gradual trend of reduction in nitrate concentration was observed at this depth until
March 2014. A close examination of the nitrate concentrations at the 1 m depth
indicated repeating fluctuations, high nitrate concentrations after harvest times due to
application of the dairy slurry, and then a reduction in concentrations. Although hard
to notice at the illustrated scale, the nitrate concentrations between September 2009
and September 2010 were still relatively high and fluctuated around ~600 mg/L (Fig.
3a). Then they escalated to ~3200 mg/L after cultivation of the pea crop. Following
this tremendous increase in nitrate concentration in May 2011, a decline was observed
until January 2012, to ~1500 mg/L (Fig. 3a). This phenomenon repeated itself in April
2012, when the nitrate concentration increased again to 2800 mg/L and then decreased
to the lower value of 78 mg/L in April 2015 due to cessation of slurry application.
Thus, application of dairy farm slurry combined with a legume crop (pea) seemed to
have enriched the top soil with excess nitrogen, compared to cultivation of cereal-type
crops.

Progression of the nitrate migration across deeper parts of the vadose zone

could be divided into two periods (Fig. 3). In the first period, October 2010 to January
2013, at depths of 2.7 m, 4.2 m, 9.5 m and 15.6 m (Fig. 3b,c,e,g), the escalation in
nitrate concentration was moderate and continuous, whereas at depths of 6.3 m and 18
m, there was no significant change in nitrate concentrations during this period (Fig.
3d,h). In the second period starting from July 2013, following the wet season of





2012/13, substantial nitrate breakthroughs were noticeable throughout most of the
vadose zone cross section (marked with arrows on Fig. 3). This rapid nitrate
progression to the deeper parts of the vadose zone could be related to the soil's
physical characteristics. In the top 3 m, the soil domain is comprised of fine-textured
layers (sandy-loam and loamy sand), and from 3 m down to 18 m (water table), the
soil consists of a coarser sand-textured layer (Turkeltaub et al., 2014). Thus, as a
consequence of substantial water percolation, which induced intensive water flux
across the coarse-textured soil, nitrate transport could be detected at deeper depths of
the vadose zone.

**3.2  Nitrate sources**

Nitrate isotope composition in the vadose zone pore water depends on nitrogen

sources and transformation processes (Böhlk, 2002). Moreover, it can provide an
indication of downward transport rates within the vadose zone (Turkeltaub et al.,
2015b). Nitrogen isotopic signature analyses were conducted on water samples
extracted by the VSPs (Fig. 4). The $\delta^{15}$N values clearly showed that manure is the
main source of nitrate in the vadose zone pore water (Fig. 4). Moreover, these values
suggested that transformation processes such as nitrification and mineralization of soil
nitrogen sources have little effect on nitrate isotopic signature. Nitrate seems to
behave like a conservative ion in most parts of the vadose zone and there is relatively
rapid nitrate transport downward to deeper parts of the vadose zone, in agreement
with previous studies (Onsoy et al., 2005; Green et al., 2008). Ultimately, nitrate
concentration time series and water content measurements from multiple depths in the



deep vadose zone demonstrated a leaching process and migration of a nitrate plume
from the soil surface toward the groundwater.

**3.3  Nitrate storage in the vadose zone**

Total nitrate mass in the unsaturated zone (as calculated by equation 2)

indicates the nitrate's fate in the vadose zone under the crop field (Fig. 5). The mass
calculation is based on the concentration time series from eight points across the
unsaturated zone. Initially, the yearly nitrate mass calculations displayed a drastic
increase in 2010, at the same time as its identification in the upper part of the vadose
zone (Fig. 3a). Subsequently, the highest increase in nitrate mass was calculated for
2011 following the combination of cultivation of a legume crop type and excessive
application of dairy slurry (Fig. 5). It seems that the yearly fluctuations in calculated
nitrate mass can be explained by the lag time in the transport process between the
sampling points. Hence, the peak in nitrate mass observed in the upper parts during
2011 (Fig. 5) remained in the vadose cross section and reached the deeper parts of the
vadose zone as a breakthrough.

**3.4  Validity of the transport model**

To verify that the observed dynamics of the nitrate concentrations compare

with earlier numerical model simulation results (Turkeltaub et al., 2014), the pore-
water velocity and the hydrodynamic dispersion were inversely estimated by solving
the analytical solution for the CDE (equation 1) (Fig. 6). The observed nitrate
concentrations at the 4.2 m depth (Fig. 3c) were used as the nitrate source and input to



the model in multi-pulse manner, and the CDE was calibrated according to nitrate
concentrations obtained at the 6.3 m depth (Fig. 3d). Close examination of the results
indicated relatively good agreement between observed and simulated nitrate
concentration trends (Fig. 6). Nevertheless there were discrepancies in the absolute
values, with the simulated nitrate concentrations increasing before the observed
concentrations. These gaps could be explained by the assumptions that are intrinsic to
the CDE model—homogeneous medium and average velocity—along with the
assumption of even distribution of the nitrogen source on the surface. However, none
of these assumptions could be found in the field. Nevertheless, the CDE provided an
approximation that could be compared with earlier numerical modeling results. The
calculated hydrodynamic dispersion was 50 cm$^2$/day and the pore-water velocity was
0.915 cm/day, which is ~333.97 cm/year. Multiplying the velocity by the average
water content observed at 3.1 m, ~0.075 cm$^3$/cm$^3$ (Fig. 2c), the Darcian flux equaled
~25 ± 9 cm/year, which is an underestimation of the earlier average flux estimation of
19.9 cm/year averaged over 24 years (Turkeltaub et al., 2014).

**3.5 Practical implications of vadose-zone monitoring**

There is much evidence indicating that agricultural practices have led to
alterations in unsaturated sediment pore-water chemicals. Moreover, agricultural land
use is one of the major non-point sources for groundwater contamination, in particular
contamination by nitrate. To prevent a long-term gradual degradation in groundwater
quality, the link between sources of pollution on the surface and their migration
pattern in the unsaturated zone should be understood long before their final
cumulative imprint in the aquifer water. Today's wide use of nitrate detection in



groundwater by standard traditional monitoring wells might be misleading due to the
mixing of waters from uncultivated and cultivated areas, which results in lower nitrate
concentrations and masks the pollution process. Hence, protection of groundwater
from potential pollution originating from agricultural land uses has to include
effective and continuous monitoring of the vadose zone. In this way, pollution events
can be monitored in their early stages, long before large-scale nitrate contamination of
the groundwater becomes inevitable.
Lastly, the VMS presented here was installed under a crop field which was
fertilized by the distribution of dairy slurry, a method that is commonly used
worldwide. This method, and its potential impact on groundwater contamination, have
been previously investigated (e.g. Basnet et al., 2001; Olson et al., 2009; Salazar et
al., 2012). However, the results presented here from continuous monitoring of the
vadose zone's hydraulic and chemical properties leave no doubt that this method
causes groundwater pollution.

**4 Conclusions**
• Application of a VMS under an agricultural field enabled real-time tracking of
water flow and nitrate transport from the surface through the entire deep
vadose zone to the water table at 18 m depth.
• The leaching process and migration of a nitrate plume were demonstrated by
nitrate concentration time series and water-content measurements from
multiple depths in the deep vadose zone underlying a crop field fertilized by
dairy slurry application.
• Isotopic composition of nitrate in the water samples indicated that manure is
the main nitrogen source for nitrate in the vadose-zone pore water. Nitrogen



transformation processes such as nitrification and mineralization seem to have
only little effect under an intensively fertilized crop field.
• Total nitrate mass estimations and simulated pore-water velocity using the
analytical solution of the convection–dispersion equation indicated dominance
of vertical nitrate transport.
• Protection of groundwater from potential pollution originating from
agricultural land uses has to include effective and continuous monitoring of
the vadose zone. Pollution events can be monitored in their early stages, long
before pollution accumulates in the aquifer water.




**Acknowledgements**

This work was funded by the Israel Water Authority (#4500687174). Thanks
go to Sara Elchanani and the Division of Water Quality of the Israel Water Authority
for supporting and funding the project. We wish to express our gratitude to the
farmers who allowed us to conduct this study in their fields. In addition, we would
like to express our appreciation to Michael Kogel for his extensive effort in
maintaining and operating the VMS. Data can be obtained by contacting the
corresponding author.





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






**Table 1**
Depth distribution of the vadose-zone monitoring system (VMS) units.

| Vertical depth from land surface (m) | |
| --- | --- |
| VSP[1,2] | FTDR[1,3] |
| 1 | 0.5 |
| 2.7 | 2.1 |
| 4 | 3.1 |
| 6.3 | 5.7 |
| 9.5 | 8.9 |
| 12.6 | 12 |
| 15.7 | 15.1 |
| 18 | 17.4 |

[1] Depth measured relative to land surface at the site.
[2] Vadose zone pore-water sampling port.
[3] Flexible time-domain reflectometry probe.




Figures

**Figure. 1.** Crop field site with monitoring location during two periods: crop growth
during the wet season (a), and after harvesting and during slurry application (b). (c)
Schematic illustration of the vadose-zone monitoring system installed under the crop
field. VSP, vadose-zone sampling port; FTDR, flexible time-domain reflectometry
sensor.

**Figure. 2.** Daily rainfall and water-content ($\theta$) variations at different depths across the
vadose zone as monitored by the flexible time-domain reflectometry sensors.

**Figure. 3.** Daily rainfall and time series of observed nitrate ($NO_3$) concentrations
which were obtained by the vadose-zone sampling ports (VSPs) at multiple depths for
6 consecutive years.

**Figure. 4.** $\delta^{15}N$ profile of nitrate in the water samples obtained from the vadose zone
under the crop field.

**Figure. 5.** Yearly total nitrate mass of the entire vadose zone over the years of
sampling.

**Figure. 6.** Observed (red circles) and simulated (dashed blue line) nitrate
concentrations for the vadose-zone sampling port (VSP) at the 6.3 m depth. The
nitrate concentrations obtained by the VSP at the 4.2 m depth served as a multi-pulse
input boundary condition.




























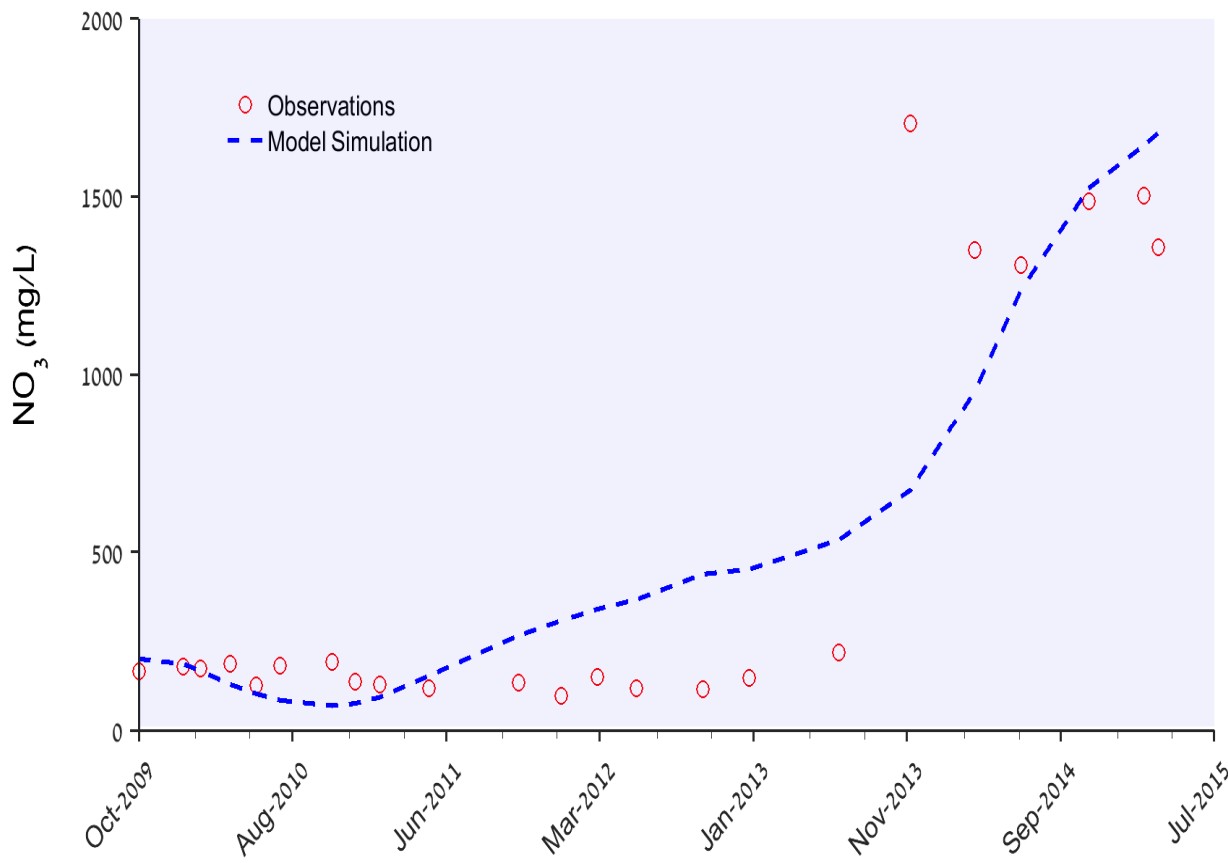