# Peer review of "Real-time monitoring of nitrate transport in deep vadose zone under a crop"

_Hydrology and Earth System Sciences, 2016_

## Referee Comment (RC1) · Anonymous Referee #1 · 12 Mar 2016

Thank you for the opportunity to review "Real-time monitoring of nitrate transport in deep vadose zone under a crop field - implications for groundwater protection." I enjoyed reading the paper. The text is clear and well written. The structure of the paper is concise and generally easy to follow. A notable strength of the paper is the dataset of 6 years of $NO_3-$ concentrations in the vadose zone, which is an unusually long and complete record. There is room for improvement in the data interpretation and discussion. Some of the interpretations of the data seem a bit oversimplified. I suggest revising sections dealing with data interpretation and discussion in order to provide readers with a more in-depth understanding of the data, and to highlight the most novel contributions of this study. Specific suggestions are given below.

[Figure]

Specific comments:

Line 30 – "and these isotopes were barely affected by natural soil or industrial nitrogen components." It is not clear how this conclusion is reached. I suggest revisiting the interpretation of the d15N results.

For Figure 4, please provide a reference for the arrows showing ranges of 15N ratios for different sources. They seem to be from Figure 16.9 in Kendall, 1998, (Tracing Nitrogen Sources and Cycling in Catchments, in Isotope Tracers in Catchment Hydrology (1998), C. Kendall and J. J. McDonnell (Eds.). Elsevier Science B.V., Amsterdam. pp. 519-576). Because there are no d18O measurements in this study, a better source of information about typical ranges of d15N in sources might be figure 16.4 in that same chapter, which shows manure separate from septic: http://wwwrcamnl.wr.usgs.gov/isoig/isopubs/itchfig16-4.html or Fogg et al., 1998, Figure 1 https://info.ngwa.org/GWOL/pdf/981563606.PDF In either case, it seems that many of the measurements shown in this figure could be from soil organic N, and that all the measurements could be mixtures of soil N and manure. I suggest revising the discussion accordingly.

Also, there is an interesting trend in the d15N values with depth (from 0 to 12.6 m), and the sample at 12.6 m has relatively low d15N. Did the source of N vary over time? Given the estimated vertical velocity of 0.9 m/yr and the date that these samples were collected, it should be possible to estimate the time at which the $NO_3-$ in the 12.6 m sample entered the soil. What was going on at this site at that time?

33 – "excluded the possibility of lateral nitrate input". It's not clear how the model results lead to this statement. How does a 1-D model, fitted to breakthrough at a single depth, "exclude" the possibility of lateral nitrate input?

88-91 – In terms of the novelty of the current study, this seems to be an important point that there are few 5+ year monitoring studies of $NO_3-$ in the deep vadose zone. Consider moving this to a more prominent location and/or adding a similar statement
in the abstract.

98 – 106 – I suggest adding a sentence to clarify how the existing studies relate to the current study. How does this study build on, or differ from, the previous VMS studies of long-term monitoring of NO3− in the unsat zone, especially those at the same site (Turkeltaub et al., 2014)? This will help to clarify the novelty of the current study.

122 – Please add brief info about samples taken for N and for d15N. How were the samples collected? When were the d15N samples collected?

126 – Please give the start and stop dates of the study in this sentence or in the first sentence of the paragraph below (line 136).

136 – Please explain - what was the land use of the study site before 2009?

320 – Meaning is unclear for "none of these assumptions could be found in the field". Presumably this means that the assumptions were not violated, but that seems inconsistent with figures showing NO3− breakthrough that is not consistent with a uniform homogeneous medium, e.g. in 2013 there is breakthrough at depths of 9.5m and 15.6m but not at the intervening port at 12.6m. I suspect that the 1-D analysis approach would not do a good job of fitting all the depths simultaneously, and that different sample depths/locations have different effective transport properties. Consider discussing this issue in more detail, relating to other unsat zone studies, and possibly proposing measures to address the related uncertainty.

326-327 – Please clarify - The model was calibrated against data from the depth of 6.3m, so for consistency it would seem to make sense to use the water content at that depth (rather than at 3.1m) to calculate the annual water flux.

329 – Section 3.5 – I suggest revising to emphasize the most novel results of this study. Some additional analysis may be necessary in previous sections to identify the most novel contributions. The first paragraph of this section is very generic, more like introductory material than discussion. The second paragraph mentions the current study,

but does not explain how the results of this study add to our knowledge of vadose zone monitoring as a tool to understand NO3− delivery to groundwater. Readers already know that applying manure slurry contaminates groundwater. They will be more interested to read about what this study (maybe in combination with previous studies) tells us about time-scales of transport, interactions of 15N with soil N, magnitude of dispersivity, or other open questions about vadose zone N.

Technical issues:

55 – "the land surface"

56-64 – Consider combining this single-sentence paragraph with another paragraph, or expand to clarify how these various methods relate to the current study. E.g. Is there a lack of studies that characterize long-term NO3− concentration profiles in the vadose zone?

77 – I suggest "concentrations" or "levels" instead of "level". Is the point of this sentence that it can advantageous to study NO3− transport in the unsaturated zone, closer to the source, before mixing occurs in groundwater?

80 – Consider substituting "common practice" in place of "easy".

108 – "patterns"

176 – should be "were analyzed"

186 – Throughout the paper consider using "vadose sampling port", "sampling port" or just "port", which will have more meaning to readers than the abbreviation "VSP".

219 – "From September 2009 to the end of the study in January 2015"?

236 – Consider giving an approximate interval between sampling (e.g. "approximately 4 times per year") in place of "frequent"

256 – "as compared to"

277 – missing "e" at end of author's name

Fig 1 – Is there any significance to the color distinction for blue versus red arrows?

Fig 3 – The different vertical scales on the panels makes it difficult to interpret the data. Consider plotting a few different depths on a single panel.

―――――――――――――――

---

## Referee Comment (RC2) · Anonymous Referee #2 · 20 Mar 2016

**General Comments**

I appreciate the opportunity to review this paper. The authors monitored the fate of nitrate in the vadose zone under an annual cropped field, which received applications of liquid dairy manure. The study was carried out for 6 years. The risk of nitrate contaminating groundwater from commercial fertilizer and organic sources is not new and has been well studied by many researchers, and therefore, this work does not add substantially in this area. However, the method used and that six years of data that were collected provides some level of novelty and strength, and may provide important regional relevance, though this is not indicated. The paper is generally well written; however, there are several places where the English is awkward and needs to be rephrased,

excess words deleted, and/or 'tighten up'. The Introduction does not state a clear purpose/objective(s)/hypothesis. Significant improvement on the description of methods is required. Many questions were raised during the review. Also, method statements appeared in the Results and Discussion and should be removed. A better description of methods and added information will improve the basis for discussion points. The area that needs the most improvement is that much more and in-depth discussion is required.

Specific Comments

In the Introduction, for several points, many references are listed. If possible, please reduce to a fewer number and most relevant references.

L107-120 Delete. This paragraph describes methods and some general results. Replace with a paragraph stating why this particular study was conducted, its importance, and the main objectives (or hypotheses).

L138 Was this tillage fallow or chemical fallow? What was the surface condition during the fallow period?

L141 What time of the year was manure applied? At what application rate (L/ha)? Did the entire field receive manure each year, or was manure applied to only a portion of the field in a given year? Do you have nutrient content data for the manure? How long has manure been applied prior to the study period? Is there any indication that manure had been 'over applied' relative to crop requirements? For example, is plant-available P high or low in the top 15 cm of soil? These are very important details for the discussion.

L148-149 Move the first part of this sentence to Section 2.1. Replace the rest of the sentence with 'The field was instrumented with a VMS (Fig. 1).' State when the instrumentation was installed

L153 Is this 35 degree from vertical or from horizontal?

L154 Boreholes is plural, suggesting more than one borehole. However, there is no further indication if there was more than one borehole. Please make it clear on the number of boreholes/VMSs. Also state where the borehole(s) was(were) installed within the field. If only one borehole was used, the study would have been strengthen if more than one was installed. Provide statements on how representative the selected borehole site was of the field.

L165-168. Delete. This is redundant.

L171 How was water content monitored? Were the FTDRs connected to dataloggers? If so, what type and how were they powered?

L172 How was water samples collected and processed? Were the VSPs connected to tubing and the water pumped to the surface? How much water was collected per sampling? How were the water samples handled in the field (e.g., placed on ice) and transported to the lab? How were the samples stored/preserved prior to analysis? What parameters were analyzed and what methods were used (with references)? Indicate the time period water samples were collected (e.g., from 2009 to 2015).

L214-215 Delete the first sentence. It is a methods statement.

L230-234 Delete. Should be in the Methods section.

L235-237 Delete the first sentence. Redundant. Already stated in the Methods section.

L252-254 Was manure applied after the wheat crop in 2013? And if so, why was no NO3 spike observed. It would be helpful to clearly state (and even show with an arrow in Fig. 3) when manure was last applied.

L255-257 This discussion needs to be expanded here. The quality and rates of the manure used at the site would be very helpful. Also the mechanism of how legumes contribute to the increase in nitrate should be discussed with references. Can specific information about the total residue biomass of the pea crop and the likely TN contribution be included?

L269-272 The mechanism/progress should be expanded and further discussed with references.

L280 The isotope analysis needs to be mentioned and described in the Methods.

L284 Often nitrate is not considered as a conservative tracer, for example, compared to chloride. Provide further discussion in this paragraph, with references. Are there variations amount studies and soil types? How does your field site/soil type compare?

L293-296 Delete the first two sentences.

L299 What basis is the application rate considered "excessive"? There is no information provided to support this.

L302-304 Therefore, essentially most of the increased NO3 remained with the vadose zone within the time frame of the study. Any speculation on when or how much of this NO3 will enter the groundwater? Please provide discussion. What are the risks?

L308-314. Delete these two sentences. They are method statements.

L326. You state there was an "underestimation". So why the difference? Please discuss.

L349-351 This statement is far too generalized. Under the conditions of the site, this is true. However, some important conditions for this site have not been described, such as the amount and quality of the manure applied. Are nutrients being over applied? But this may not be the case at other sites because of a host of factors. Therefore, this needs to be re-phrased along with further discussion. In areas that are at higher risk of groundwater contamination from nitrogen sources, particularly from manure, what mitigation options are potentially available? There should be some discussion around this. For example, apply manure based on crop requirements (e.g., see Olson et al. 2010. Canadian Journal of Soil Science 90; 619-635).

L354-360 The first two conclusion points are essentially the same thing. Please com-

bine.

L363-364 Provide supporting discussion as to why nitrification and mineralization had little effect at this site. Discuss.

L368-371 This is not a methods paper. I assume this is a proven method to monitor leaching of contaminant and water content in the vadose zone. Instead, state what are potential mitigation options, future work required, other practical implications, etc. Is there a local/regional significance to this work?

L380-381 This implies more than one field. However, the Methods/Results suggests that only one field was used in the study. This adds more uncertainly on what was actually done in the study.

Technical Corrections

L23 Replace 'over a period of' with 'for'

L24 delete 'deep'

L25 delete 'sediment'

L45 add 'as' before NO3, and (WHO, 2011) after NO3. The reference is WHO 2011 4th edn.

L50 Units should be written exponentially mg L-1. Applies throughout the paper.

L53 Change 'mechanism' to 'mechanisms'

L54 After the word 'specific', replace the rest of the sentence with 'practices used on agricultural land'

L57 Delete the colon

L57 Add the word 'analysis' after signature

L67 Replace 'evolve' with 'change'

L75 After 'Therefore,' add 'our understanding of'

L76 Replace 'impact' with 'effect'

L81 Replace 'water' with 'as a source for drinking water'

L83 Replace 'impact' with 'effect'

L85 Replace 'which' with 'that'

L86 Replace 'impact' with 'effect'

L86 Delete 'the' at the end of the line

L87 There is no Scanlon et al. 2002 in the list of references. Possibly this should be 2010.

L89 Replace 'over' with 'during'

L98 Replace 'domain' with 'zone'

L100 Replace 'setups' with 'settings'

L100 Delete the colon

L105 Replace 'impact' with 'effect'

L126 Delete 'located'

L129 add 'with' before 'an' near the end of the line

L131 change 'month' to 'months'

L138 delete the comma after 'Then'

L138-140 Delete 'with no additional irrigation' It has already been stated that this is a rainfed site.

L141 After harvest, the field was plowed with a . . . (described/name the implement).

L141 Delete 'crop'

L142 Replace 'distribution' with 'application'

L146 Delete 'setup'

L153 Add a comma after 'uncased'

L154 Replace 'multiple' with 'eight'

L155 Replace the first two words (has a) with 'consisted of a'

L156 Add 'a' before vadose-zone

L156 Change 'ports' to 'port'

L156 Change VSPs to VSP

L160 Replace 'is' with 'was'

L161 material (liquid two-component urethane), which solidified . . ...

L162 Replace 'attachment' with 'good contact'

L163 Replace 'to' with 'with'

L186 Delete 'located'

L191 Delete ', both'

L192 In the list of reference, it appears as Van.

L206 M/L3 is an odd unit. Does M represent mole? And you cannot have a cubic litre.

L216 Replace 'indicated' with 'show'

L217 Change 'contents' to 'content'

L220 Replace 'significant' with 'larger'

L226 Delete 'down'

L228 Delete 'as well'

L237-238 Delete 'different scales and magnitudes of the'

L238-239 Change the first part of the sentence so is reads, The nitrate concentration time . . .

L240 After 'surface' add 'in 2011 and 2012'

L241 Replace ∼ with 'about' Appears elsewhere in the manuscript.

L246 'with higher' and delete 'times'

L247 . . . then followed by a reduction . . .

L248 . . . scale in Fig. 3a, . . ..

L249-250 . . .fluctuated neat 600 mg/L. Then concentration increased to about 32000 mg/L . . ..

L251 Replace 'tremendous' with 'relatively large'

L252 Delete the comma and change ∼ to 'about'

L254 Delete 'the lower value of'

L258 . . .. migration deeper into the vadose . . ..

L259 Replace 'could' with 'can'

L259 Delete (Fig. 3)

L260 . . . of 2.7, 4.2, 9.5, and 15.6 m . . ...

L260 Replace 'escalation' with 'increase'

L261 Change the comma to a semi-colon, add a comma after 'whereas' and delete m

L262 Replace 'significant' with 'major'

L262 Delete 'during this period'

L263 Add a comma after 'period' and delete the comma after '2013'

L265 Replace 'on' with 'in'

L267 Delete 'domain is'

L268 Delete 'm down'

L269 Replace 'consists' with 'consisted'

L279-280 Delete this sentence Nitrogen . . .. (Fig. 4).

L284 Replace 'like' with 'as'

L286 Replace the first half of the sentence. At the study site, measurements showed leaching and migration of a . . .

L296 Add '(Eq. 2) after 'calculations'

L296 Replace 'a drastic' with 'an'

L297 . . . increase from 2009 to 2010, at the same time as NO3 concentration increased in the upper . . .

L299 . . . cultivation of the pea crop and excessive . . .

L314 Delete 'Close examination of the' The results . . ...

L319 After 'model' add '(Eq. 1)'

L321 Replace 'found in' with 'applied to'

L324-326 Replace $\sim$ with 'about'

L327 Replace 'over' with 'for'

L336-337 Delete the last part of the sentence after 'understood'

[Figure]

L337 Replace 'Today's' with 'The'

L338 Replace 'might' with 'may'

L345 Replace 'which' with 'that'

L346 Replace 'by the distribution' with 'with'

L347 Replace 'impact' with 'effects'

L540 Delete the first footnote. It is not needed as the heading in the table already indicates this.

L546-549 Show Fig. 1c as a separate diagram. The diagram shows an observational well. There is no mention of this well in the methods or elsewhere in the paper. Please remove from the diagram. Show a distance scale in the diagram to indicate that the water table is about 18 m below the soil surface.

L551-552 Figure 2. Water-content (o) at different depths in the vadose zone and daily rainfall for six consecutive years.

L554-556 Figure 3. Time series of observed (NO3) concentrations in the vadose zone and daily rainfall for six consecutive years.

L561-562 . . .. entire vadose zone per year.

L564 Delete '(red circle)' and '(dashed blue line)' Figs. 2 and 3 The text in the these two figures seem to be stretched. Please re-size the figures.

Note: All of the citations in the list of references appeared in the text.

END

---

## Author Comment (AC1) · 10 May 2016

We would like to thank the reviewer for his/her quick and helpful review. The comments were constructive and helped us to improve our manuscript. All comments were addressed in reply to reviewer comments.

Specific comments: Comment 1: Line 30 – "and these isotopes were barely affected by natural soil or industrial nitrogen components." It is not clear how this conclusion is reached. I suggest revisiting the interpretation of the d15N results. Reply to comment

1: The reviewer's comment was accepted and the manuscript was revised accordingly (Lines 27-29). To strength up the result and show robustness of our conclusions we added to figure 4 ïĂăïĄď18O VS ïĄď15N measurements. The ïĂăïĄď18O -ïĂăïĄď15N isotopic composition method was used in previous investigations to assess sources of nitrate in groundwater (e.g. Wassenaar 1995), and in vadose zone as well (e.g. Turkeltaub et al., 2015b). According to Figure 4b, all water samples are situated within the manure range.

Turkeltaub, T., Kurtzman, D., Russak, E. E., and Dahan, O.: Impact of switching crop type on water and solute fluxes in deep vadose zone, Water Resour. Res., 51, 9828–9842, doi:10.1002/2015WR017612, 2015b. Wassenaar, L. I.: Evaluation of the origin and fate of nitrate in the Abbotsford Aquifer using the isotopes of 15N and 18O in NO3, Appl. Geochem., 10, 391–405, doi:10.1016/0883-2927(95)00013-A, 1995.

Comment 2: For Figure 4, please provide a reference for the arrows showing ranges of 15N ratios for different sources. They seem to be from Figure 16.9 in Kendall, 1998, (Tracing Nitrogen Sources and Cycling in Catchments, in Isotope Tracers in Catchment Hydrology (1998), C. Kendall and J. J. McDonnell (Eds.). Elsevier Science B.V., Amsterdam. pp. 519-576). Because there are no d18O measurements in this study, a better source of information about typical ranges of d15N in sources might be figure 16.4 in that same chapter, which shows manure separate from septic: http://wwwrcamnl.wr.usgs.gov/isoig/isopubs/itchfig16-4.html or Fogg et al., 1998, Figure 1 https://info.ngwa.org/GWOL/pdf/981563606.PDF In either case, it seems that many of the measurements shown in this figure could be from soil organic N, and that all the measurements could be mixtures of soil N and manure. I suggest revising the discussion accordingly. Reply to comment 2: The reviewer's comment was accepted, figure 4 and the manuscript were revised accordingly (Line 295; Lines 513 - 514). Another graph was added to figure 4, which displays the ïĂăïĄď18O measurements as well. The discussion was not revised, since it was based on the analysis following the Dual Nitrate Isotopes method.

Comment 3: Also, there is an interesting trend in the d15N values with depth (from 0 to 12.6 m), and the sample at 12.6 m has relatively low d15N. Did the source of N vary over time? Given the estimated vertical velocity of 0.9 m/yr and the date that these samples were collected, it should be possible to estimate the time at which the $NO_3^-$ in the 12.6 m sample entered the soil. What was going on at this site at that time? Reply to comment 3: As the reviewer mentioned, the water sampling obtained at 12.6 m depth is different and diverts from the general trend displayed by the rest of the water samples obtained by the sampling ports. This might be explained by the spatial variability which the vadose zone monitoring system captures. Each monitoring unit faces a different undisturbed sediment column that extends from land surface to the probe or sampling port depth. Therefore, the patterns of the measured processes as water percolation and nitrate transport might be somewhat different for each sampling port or water sensor. However, earlier studies with the vadose zone monitoring system measurements indicate that 1D vertical analysis could explain significant processes which occur in the deep vadose zone. A new estimation of the pore water velocity (see reply to comment 10 and lines 341-349), after including the nitrate time series obtained at the 9.5, 15.3 and 18 m depths, is 3.05 m/y. The nitrate origin from manure source (dairy slurry) reached to 12.6 m depth after 3.8 - 4.1 years, which is in the range of the current study.

Comment 4: 33 – "excluded the possibility of lateral nitrate input". It's not clear how the model results lead to this statement. How does a 1-D model, fitted to breakthrough at a single depth, "exclude" the possibility of lateral nitrate input? Reply to comment 4: We accepted the reviewer's comment that the model results and the total nitrate mass calculation could not exclude lateral nitrate input and revised the manuscript accordingly (Line 33). Generally, the driving forces for vertical flow component in the unsaturated zone are significantly higher compared to the lateral vectors. Moreover, there are no other potential sources of nitrate and water on surface in the vicinity of the monitored field. Hence it is likely that lateral contribution of nitrate is negligible. The vertical pore-water velocity was estimated with 1-D transport model. Subsequently, the velocity value was compared with results from earlier study on the same site, Turkeltaub et al. (2014), where the recharge fluxes were estimated with 1-D flow transient numerical model. The models were calibrated to different data sets. In Turkeltaub et al. (2014) the flow model was constrained to climate data as top boundary condition and calibrated to water content measurements obtained from multiple depths of the deep vadose zone. The data used for the 1D model in the current study were the nitrate-concentration time series. Nevertheless, both models estimated similar velocities. We would expect that any lateral nitrate or water contribution would affect the models results, which did not occur. Moreover, the total nitrate mass calculations indicated that the nitrate mass observed in the upper parts, remained in the vadose zone cross section and reached the deeper parts of the vadose zone.

Comment 5: 88-91 – In terms of the novelty of the current study, this seems to be an important point that there are few 5+ year monitoring studies of NO3− in the deep vadose zone. Consider moving this to a more prominent location and/or adding a similar statement in the abstract. Reply to comment 5: The reviewer's comment was accepted and the manuscript was revised accordingly (Lines 23 - 24). Furthermore, the Introduction section was reorganized and some parts were omitted (Lines 42 -106).

Comment 6: 98 – 106 – I suggest adding a sentence to clarify how the existing studies relate to the current study. How does this study build on, or differ from, the previous VMS studies of long-term monitoring of NO3− in the unsat zone, especially those at the same site (Turkeltaub et al., 2014)? This will help to clarify the novelty of the current study. Reply to comment 6: The reviewer's comment was accepted and the manuscript was revised accordingly (Lines 83-94).

Comment 7: 122 – Please add brief info about samples taken for N and for d15N. How were the samples collected? When were the d15N samples collected? Reply to comment 7: The reviewer's comment was accepted and the manuscript was revised accordingly (Lines 160 - 173).

[Figure]

Comment 8: 126 – Please give the start and stop dates of the study in this sentence or in the first sentence of the paragraph below (line 136). Reply to Comment 8: The reviewer's comment was accepted and the manuscript was revised accordingly (Line 117).

Comment 9: 136 – Please explain - what was the land use of the study site before 2009? Reply to Comment 9: The reviewer's comment was accepted and the manuscript was revised accordingly (Line 125-127).

Comment 10: 320 – Meaning is unclear for "none of these assumptions could be found in the field". Presumably this means that the assumptions were not violated, but that seems inconsistent with figures showing $NO3-$ breakthrough that is not consistent with a uniform homogeneous medium, e.g. in 2013 there is breakthrough at depths of 9.5m and 15.6m but not at the intervening port at 12.6m. I suspect that the 1-D analysis approach would not do a good job of fitting all the depths simultaneously, and that different sample depths/locations have different effective transport properties. Consider discussing this issue in more detail, relating to other unsat zone studies, and possibly proposing measures to address the related uncertainty. Reply to comment 10: We intended to explain the discrepancies between observed and simulated values, and indicated that the gaps might be a product of the analytical solution assumptions. However, this sentence was misleading and therefore was deleted (Line 334). The nitrate time series from the 9.5, 15.6 and 18 m depths were included in the calibration efforts. Although there is a relatively good agreement between simulated and observed nitrate concentrations, there are still discrepancies that related to the CDE assumption of steady average velocity and homogeneous medium. Obviously, the analytical solution does not stand alone and it is compared with earlier numerical simulation efforts. Moreover, we added a discussion concerning the calculated effective dispersivity coefficient. We agree with the reviewer's comment that there are horizontal and vertical distributions of the effective transport properties in the soil. However, comparison between the observations from all depths indicated that only the observations obtained

at the 12.6 m depth divert from the general observed trend. Yet, since each and every monitoring unit is located under a different soil profile, we relate this deviation to spatial variability that was not considered in this work. It can be investigated only with 3d model simulations, which are not in the scope of this work.

Comment 11: 326-327 – Please clarify - The model was calibrated against data from the depth of 6.3m, so for consistency it would seem to make sense to use the water content at that depth (rather than at 3.1m) to calculate the annual water flux. Reply to comment 11: The reviewer's comment was accepted and the manuscript was revised accordingly (Line 341). Since we used water content observations from different depths within the sandy layer of the vadose zone, weighted average water content was calculated.

Comment 12: 329 – Section 3.5 – I suggest revising to emphasize the most novel results of this study. Some additional analysis may be necessary in previous sections to identify the most novel contributions. The first paragraph of this section is very generic, more like introductory material than discussion. The second paragraph mentions the current study, but does not explain how the results of this study add to our knowledge of vadose zone monitoring as a tool to understand $NO_3-$ delivery to groundwater. Readers already know that applying manure slurry contaminates groundwater. They will be more interested to read about what this study (maybe in combination with previous studies) tells us about time-scales of transport, interactions of 15N with soil N, magnitude of dispersivity, or other open questions about vadose zone N. Reply to comment 12: The reviewer's comment was accepted. We revised section 3.5 - 'Practical implications of vadose-zone monitoring', since both reviewers indicated that this section should emphasize and elucidate the novelty of this study (Lines 359-391). Technical issues: Comment 13: Comment 55 – "the land surface" Reply to comment 13: The reviewer's comment was accepted. However, we rephrase the last sentence of this paragraph and these words were excluded from the manuscript (Line 54).

Comment 14: 56-64 – Consider combining this single-sentence paragraph with another

paragraph, or expand to clarify how these various methods relate to the current study. E.g. Is there a lack of studies that characterize long-term NO3− concentration profiles in the vadose zone? Reply to comment 14: The reviewer's comment was accepted, the Introduction section was reorganized and some parts were omitted (Lines 55 -67).

Comment 15: 77 – I suggest "concentrations" or "levels" instead of "level". Is the point of this sentence that it can advantageous to study NO3− transport in the unsaturated zone, closer to the source, before mixing occurs in groundwater? Reply to comment 15: The reviewer's comment was accepted and the manuscript was revised accordingly (Line 77).

Comment 16: 80 – Consider substituting "common practice" in place of "easy". Reply to comment 16: The reviewer's comment was accepted, however since the Introduction section was reorganized this sentence was deleted (Lines 42 -106).

Comment 17: 108 – "patterns" Reply to comment 17: The reviewer's comment was accepted and the manuscript was revised accordingly (Line 97).

Comment 18: 176 – should be "were analyzed" Reply to comment 18: The reviewer's comment was accepted and the manuscript was revised accordingly (Line 185).

Comment 19: 186 – Throughout the paper consider using "vadose sampling port", "sampling port" or just "port", which will have more meaning to readers than the abbreviation "VSP". Reply to comment 19: The reviewer's comment was accepted and the manuscript was revised accordingly (Lines 147, 150, 161, 163, 196, 213, 214, 241, 295; table 1 and figure captions of figures 1 and 6).

Comment 20: 219 – "From September 2009 to the end of the study in January 2015"? Reply to comment 20: The reviewer's comment was accepted; however the sentence was deleted following the second reviewer's comment (Line 222).

Comment 21: 236 – Consider giving an approximate interval between sampling (e.g. "approximately 4 times per year") in place of "frequent" Reply to comment 21: The

reviewer's comment was accepted; however the sentence was deleted following the second reviewer's comment (Line 236).

Comment 22: 256 – "as compared to" Reply to comment 22: The reviewer's comment was accepted and the manuscript was revised accordingly (Line 267).

Comment 23: 277 – missing "e" at end of author's name Reply to comment 23: The reviewer's comment was accepted and the manuscript was revised accordingly (Line 292).

Comment 24: Fig 1 – Is there any significance to the color distinction for blue versus red arrows? Reply to comment 24: We accepted the reviewer's comment and Figure 1 was redesigned. All arrows were painted with blue color to avoid any misunderstanding. Comment 25: Fig 3 – The different vertical scales on the panels makes it difficult to interpret the data. Consider plotting a few different depths on a single panel. Reply to comment 25: We accepted the reviewer's comment and Figure 3 was redesigned.

---

## Author Comment (AC2) · 10 May 2016

We would like to thank the reviewer for his/her quick and helpful review. The comments were constructive and helped us to improve our manuscript. All comments were addressed in reply to reviewer comments

Specific Comments:

Comment 1: In the Introduction, for several points, many references are listed. If possible, please reduce to a fewer number and most relevant references.

Reply to comment 1: The reviewer's comment was accepted and the manuscript was revised accordingly. The Introduction section was reorganized and some parts were omitted (Lines 42 -106).

Comment 2: L107-120 Delete. This paragraph describes methods and some general results. Replace with a paragraph stating why this particular study was conducted, its importance, and the main objectives (or hypotheses).

Reply to comment 2: The reviewer's comment was accepted and the manuscript was revised accordingly. The Introduction section was reorganized (Lines 42 -106).

Comment 3: L138 Was this tillage fallow or chemical fallow? What was the surface condition during the fallow period?

Reply to comment 3: The reviewer's comment was accepted and the manuscript was revised accordingly (Line129). Figure 2 and Figure 3 were revised as well. The implication here referred to the fact that the field was not cultivated for a period of a year time.

Comment 4: L141 What time of the year was manure applied? At what application rate (L/ha)? Did the entire field receive manure each year, or was manure applied to only a portion of the field in a given year? Do you have nutrient content data for the manure? How long has manure been applied prior to the study period? Is there any indication that manure had been 'over applied' relative to crop requirements? For example, is plant-available P high or low in the top 15 cm of soil? These are very important details for the discussion.

Reply to comment 4: The reviewer's comment was accepted and all details known to us by personal communication with the farmers were added to the manuscript (Line 125-129). The dairy farmer has a limited period of time during the year (May and June) to dispose the dairy wastes by distribution them over the field. However, to date, there

is no limit for application of slurry which the farmer is constrained to. Therefore in many occasions, the method ignores the crop demand. For example, the slurry was distributed after the pea crop cultivation, which already enriched the soil with nitrogen (∼86 Kg/ha, Herridge et al., 2008), and given that the recommended fertilization application for wheat according to the Agriculture Extension Service of Israel is between 40 and 100 kg/ha. Therefore, the manure was defiantly over applied and consequence in nitrate leaching beyond root zone.

Herridge, D., Peoples, M. and Boddey, R.: Global inputs of biological nitrogen fixation in agricultural systems, Plant Soil, 311, 1–18, doi:10.1007/s11104-008-9668-3, 2008.

Comment 5: L148-149 Move the first part of this sentence to Section 2.1. Replace the rest of the sentence with 'The field was instrumented with a VMS (Fig. 1).' State when the instrumentation was installed

Reply to comment 5: The reviewer's comment was accepted and the manuscript was revised accordingly (Lines 112-113 and line 140).

Comment 6: L153 Is this 35 degree from vertical or from horizontal?

Reply to comment 6: The reviewer's comment was accepted and the manuscript was revised accordingly (Line 145). The slanted borehole is 350 from the vertical.

Comment 7: L154 Boreholes is plural, suggesting more than one borehole. However, there is no further indication if there was more than one borehole. Please make it clear on the number of boreholes/VMSs. Also state where the borehole(s) was(were) installed within the field. If only one borehole was used, the study would have been strengthening if more than one was installed. Provide statements on how representative the selected borehole site was of the field.

Reply to comment 7: We apologize for the mistake. Although, originaly two VMSs were installed only one of the systems was fully functioning. The reviewer's comment was accepted and the manuscript was revised accordingly (Line 145).

The VMS installed under the crop field is part of an array of VMSs that were installed under different representative land-uses situated above the southern part of the Israeli costal aquifer (Dahan et al., 2014, Baram et al., 2013, 2014, Turkeltaub et al., 2014,2015a, 2015b). An investigation of each site and its findings are combined with the other studies to generate a comprehensive perspective on dominant factors controlling groundwater quality and quantities. Subsequently, these inferences could serve as guiding principles for any water-resources management decision.

Comment 8: L165-168. Delete. This is redundant.

Reply to comment 8: The reviewer's comment was accepted and the manuscript was revised accordingly (Line 157).

Comment 9: L171 How was water content monitored? Were the FTDRs connected to dataloggers? If so, what type and how were they powered?

Reply to comment 9: The reviewer's comment was accepted and the details were added to the manuscript (Lines 170-173).

Comment 10: L172 How was water samples collected and processed? Were the VSPs connected to tubing and the water pumped to the surface? How much water was collected per sampling? How were the water samples handled in the field (e.g., placed on ice) and transported to the lab? How were the samples stored/preserved prior to analysis? What parameters were analyzed and what methods were used (with references)? Indicate the time period water samples were collected (e.g., from 2009 to 2015).

Reply to comment 10: The reviewer's comment was accepted and the details were added to the manuscript (Lines 160-170).

Comment 11: L214-215 Delete the first sentence. It is a methods statement.

Reply to comment 11: The reviewer's comment was accepted and the manuscript was revised accordingly (Lines 222).

Comment 12: L230-234 Delete. Should be in the Methods section.

Reply to comment 12: The reviewer's comment was accepted and the manuscript was revised accordingly (Line 236).

Comment 13: L235-237 Delete the first sentence. Redundant. Already stated in the Methods section.

Reply to comment 13: The reviewer's comment was accepted and the manuscript was revised accordingly (Line 236).

Comment 14: L252-254 Was manure applied after the wheat crop in 2013? And if so, why was no NO3 spike observed. It would be helpful to clearly state (and even show with an arrow in Fig. 3) when manure was last applied.

Reply to comment 14: The manure wasn't applied to the field in May 2013, since the farmer decided to plant jojoba (Simmondsia chinensis) shrubs (personal communication). Although it took another year till plantation occurred. A solid line arrow was drawn to show the last manure application time in Fig. 3a.

Comment 15: L255-257 This discussion needs to be expanded here. The quality and rates of the manure used at the site would be very helpful. Also the mechanism of how legumes contribute to the increase in nitrate should be discussed with references. Can specific information about the total residue biomass of the pea crop and the likely TN contribution be included?

Reply to comment 15: There are advantages and disadvantages in studying commercial agriculture sites. Observations obtained under commercial conditions are an outcome of farming which constrained to economical and other necessities of the farmer. Therefore the observations represent the prevailed conditions over part of the coastal aquifer and the unsaturated zone in a realistic manner. However, the drawback is that the data are not always available and in many cases are approximated; especially when contamination is in risk (the farmer provided data based on good will rather than

obligation). The coarse estimations of manure application rates were added to the manuscript. Information concerning nitrogen fixation of pea crop was taken from the literature (Lines 255-266).

Comment 16: L269-272 The mechanism/progress should be expanded and further discussed with references.

Reply to comment 16: The reviewer's comment was accepted and the manuscript was revised accordingly (Lines 283-287; Lines 519-521).

Comment 17: L280 The isotope analysis needs to be mentioned and described in the Methods.

Reply to comment 17: The reviewer's comment was accepted and the manuscript was revised accordingly (Lines 175-180).

Comment 18: L284 Often nitrate is not considered as a conservative tracer, for example, compared to chloride. Provide further discussion in this paragraph, with references. Are there variations amount studies and soil types? How does your field site/soil type compare?

Reply to comment 18: We want to thank the reviewer for this comment. The issue of the factors controlling nitrate fluxes to groundwater, and especially under different soil type and agricultural land use, is still under investigation. Although there are many studies indicating on insignificant of the transformation nitrogen processes in the deep unsaturated zone beyond the root zone, some other studies displayed contrast or different conclusions. This comment focused us and led us to the conclusion of the next stage in research. A holistic approach which includes all potential factors controlling nitrate fluxes to identify the dominant once. The manuscript was revised accordingly (Lines 299-312).

Comment 19: L293-296 Delete the first two sentences.

Reply to comment 19: The reviewer's comment was accepted and the manuscript was

revised accordingly (Line 315).

Comment 20: L299 What basis is the application rate considered "excessive"? There is no information provided to support this.

Reply to comment 20: See reply to comment 4 and reply to comment 15.

Comment 21: L302-304 Therefore, essentially most of the increased NO3 remained with the vadose zone within the time frame of the study. Any speculation on when or how much of this NO3 will enter the groundwater? Please provide discussion. What are the risks?

Reply to comment 21: The reviewer's comment was accepted; however the location of this discussion in the manuscript should be in the Nitrate transport model section (lines 350-356).

Comment 22: L308-314. Delete these two sentences. They are method statements.

Reply to comment 22: The reviewer's comment was accepted and the manuscript was revised accordingly (Line 326).

Comment 23: L326. You state there was an "underestimation". So why the difference? Please discuss.

Reply to comment 23: Due to the other reviewer's comment, we simulated the nitrate time series obtained from 6.3 m, 9.5 m, 15.6 m and 18 m depths, which are all located within the sandy texture layer. The calculated pore water velocity was similar to the numerical results (Lines 331-349).

Comment 24: L349-351 This statement is far too generalized. Under the conditions of the site, this is true. However, some important conditions for this site have not been described, such as the amount and quality of the manure applied. Are nutrients being over applied? But this may not be the case at other sites because of a host of factors. Therefore, this needs to be re-phrased along with further discussion. In areas

that are at higher risk of groundwater contamination from nitrogen sources, particularly from manure, what mitigation options are potentially available? There should be some discussion around this. For example, apply manure based on crop requirements (e.g., see Olson et al. 2010. Canadian Journal of Soil Science 90; 619-635).

Reply to comment 24: The reviewer's comment was accepted. We revised section 3.5 - 'Practical implications of vadose-zone monitoring', since both reviewers indicated that this section should emphasize and elucidate the novelty of this study (Lines 359-391).

Comment 25: L354-360 The first two conclusion points are essentially the same thing. Please combine.

Reply to comment 25: The conclusion section was revised (Lines 395-414).

Comment 26: L363-364 Provide supporting discussion as to why nitrification and mineralization had little effect at this site. Discuss.

Reply to comment 26: See reply to comment 18.

Comment 27: L368-371 This is not a methods paper. I assume this is a proven method to monitor leaching of contaminant and water content in the vadose zone. Instead, state what are potential mitigation options, future work required, other practical implications, etc. Is there a local/regional significance to this work?

Reply to comment 27: The conclusion and practical implementation sections were revised. We elaborated on the mitigation options and the future work, which related to the findings from the current study and other studies sites (Lines 359-414).

Comment 28: L380-381 This implies more than one field. However, the Methods/Results suggests that only one field was used in the study. This adds more uncertainly on what was actually done in the study.

Reply to comment 28: See reply to comment 7.

Technical Corrections: Comment 29: L23 Replace 'over a period of' with 'for'

Reply to comment 29: The reviewer's comment was accepted and the manuscript was revised accordingly (Line 22).

Comment 30: L24 delete 'deep'

Reply to comment 30: The reviewer's comment was accepted and the manuscript was revised accordingly (Line 24).

Comment 31: L25 delete 'sediment'

Reply to comment 31: The reviewer's comment was accepted, however the line was deleted (Line 24). Comment 32: L45 add 'as' before NO3, and (WHO, 2011) after NO3. The reference is WHO 2011 4th edn.

Reply to comment 32: The reviewer's comment was accepted and the manuscript was revised accordingly (Line 44 and Line 487).

Comment 33: L50 Units should be written exponentially mg L-1. Applies throughout the paper.

Reply to comment 33: The reviewer's comment was accepted and the manuscript was revised accordingly (Lines 41, 49-50, 123, 134, 214, 248, 249, 251, 253, 256, 259, 264, 339-343).

Comment 34: L53 Change 'mechanism' to 'mechanisms' Reply to comment 34: The reviewer's comment was accepted and the manuscript was revised accordingly (Line 52).

Comment 35: L54 After the word 'specific', replace the rest of the sentence with 'practices used on agricultural land'

Reply to comment 35: The reviewer's comment was accepted and the manuscript was revised accordingly (Lines 53-54).

Comment 36: L57 Delete the colon

[Figure]

Reply to comment 36: The reviewer's comment was accepted and the manuscript was revised accordingly (Line 56).

Comment 37: L57 Add the word 'analysis' after signature

Reply to comment 37: The reviewer's comment was accepted and the manuscript was revised accordingly (Line 56).

Comment 38: L67 Replace 'evolve' with 'change' Reply to comment 38: We omitted this part from the manuscript.

Comment 39: L75 After 'Therefore,' add 'our understanding of' Reply to comment 39: The reviewer's comment was accepted and the manuscript was revised accordingly (Line 76).

Comment 40: L76 Replace 'impact' with 'effect' Reply to comment 40: The reviewer's comment was accepted and the manuscript was revised accordingly (Line 76).

Comment 41: L81 Replace 'water' with 'as a source for drinking water' Reply to comment 41: The reviewer's comment was accepted and the manuscript was revised accordingly (Line 67).

Comment 42: L83 Replace 'impact' with 'effect'

Reply to comment 42: The reviewer's comment was accepted and the manuscript was revised accordingly (Line 81).

Comment 43: L85 Replace 'which' with 'that'

Reply to Comment 43: We omitted this part from the manuscript.

Comment 44: L86 Replace 'impact' with 'effect'

Reply to Comment 44: We omitted this part from the manuscript.

Comment 45: L86 Delete 'the' at the end of the line

Reply to Comment 45: We omitted this part from the manuscript.

Comment 46: L87 There is no Scanlon et al. 2002 in the list of references. Possibly this should be 2010.

Reply to Comment 46: We omitted this part from the manuscript.

Comment 47: L89 Replace 'over' with 'during'

Reply to Comment 47: The reviewer's comment was accepted and the manuscript was revised accordingly (Line 63).

Comment 48: L98 Replace 'domain' with 'zone'

Reply to Comment 48: The reviewer's comment was accepted and the manuscript was revised accordingly (Line 89).

Comment 49: L100 Replace 'setups' with 'settings'

Reply to comment 49: The reviewer's comment was accepted and the manuscript was revised accordingly (Line 86).

Comment 50: L100 Delete the colon

Reply to comment 50: We omitted this part from the manuscript.

Comment 51: L105 Replace 'impact' with 'effect'

Reply to comment 51: We omitted this part from the manuscript.

Comment 52: L126 Delete 'located'

Reply to comment 52: The reviewer's comment was accepted and the manuscript was revised accordingly (Line 112).

Comment 53: L129 add 'with' before 'an' near the end of the line

Reply to comment 53: The reviewer's comment was accepted and the manuscript was

revised accordingly (Line 119).

Comment 54: L131 change 'month' to 'months'

Reply to comment 54: The reviewer's comment was accepted and the manuscript was revised accordingly (Line 120).

Comment 55: L138 delete the comma after 'Then'

Reply to comment 55: The reviewer's comment was accepted and the manuscript was revised accordingly (Line 129).

Comment 56: L138-140 Delete 'with no additional irrigation' It has already been stated that this is a rainfed site.

Reply to comment 56: The reviewer's comment was accepted and the manuscript was revised accordingly (Line 131).

Comment 57: L141 After harvest, the field was plowed with a . . . (described/name the implement).

Reply to comment 57: The reviewer's comment was accepted and the manuscript was revised accordingly (Line 131).

Comment 58: L141 Delete 'crop'

Reply to comment 58: The reviewer's comment was accepted and the manuscript was revised accordingly (Line 132).

Comment 59: L142 Replace 'distribution' with 'application'

Reply to comment 59: We omitted this part from the manuscript.

Comment 60: L146 Delete 'setup'

Reply to comment 60: The reviewer's comment was accepted and the manuscript was revised accordingly (Line 138).

[Figure]

Comment 61: L153 Add a comma after 'uncased'

Reply to comment 61: The reviewer's comment was accepted and the manuscript was revised accordingly (Line 144).

Comment 62: L154 Replace 'multiple' with 'eight'

Reply to comment 62: The Vadose Zone Monitoring System could host multiple monitoring units. The number of units is defined according to the monitoring demand and the investigated vadose zone thickness. In the section where the line mentioned above, a general description about the VMS is given. Therefore we disagree to the suggested correction.

Comment 63: L155 Replace the first two words (has a) with 'consisted of a'

Reply to comment 63: The reviewer's comment was accepted and the manuscript was revised accordingly (Line 146).

Comment 64: L156 Add 'a' before vadose-zone

Reply to comment 64: The reviewer's comment was accepted and the manuscript was revised accordingly (Line 147).

Comment 65: L156 Change 'ports' to 'port'

Reply to comment 65: The reviewer's comment was accepted and the manuscript was revised accordingly (Line 147).

Comment 66: L156 Change VSPs to VSP

Reply to comment 66: We omitted this abbreviation according to the other reviewer's suggestion.

Comment 67: L160 Replace 'is' with 'was'

Reply to comment 67: The reviewer's comment was accepted and the manuscript was revised accordingly (Line 151).

Comment 68: L161 material (liquid two-component urethane), which solidified . . ...

Reply to comment 68: The reviewer's comment was accepted and the manuscript was revised accordingly (Line 152).

Comment 69: L162 Replace 'attachment' with 'good contact'

Reply to comment 69: The reviewer's comment was accepted and the manuscript was revised accordingly (Line 154).

Comment 70: L163 Replace 'to' with 'with'

Reply to comment 70: The reviewer's comment was accepted and the manuscript was revised accordingly (Line 154).

Comment 71: L186 Delete 'located' Reply to comment 71: The reviewer's comment was accepted and the manuscript was revised accordingly (Line 184).

Comment 72: L191 Delete ', both'

Reply to comment 72: The reviewer's comment was accepted and the manuscript was revised accordingly (Line 199).

Comment 73: L192 In the list of reference, it appears as Van.

Reply to comment 73: There is capital 'V' in the reference because it is the first letter in the sentence. I checked it with different papers and they all referenced 'van' with small 'v' within the text and capital 'V' within the reference section.

Comment 74: L206 M/L3 is an odd unit. Does M represent mole? And you cannot have a cubic litre.

Reply to comment 74: M represents mass and L represents length unit. This is a general writing of units. Many papers use these general signs.

Comment 75: L216 Replace 'indicated' with 'show'

Reply to comment 75: The reviewer's comment was accepted and the manuscript was revised accordingly (Line 222).

Comment 76: L217 Change 'contents' to 'content'

Reply to comment 76: The reviewer's comment was accepted and the manuscript was revised accordingly (Line 223).

Comment 77: L220 Replace 'significant' with 'larger'

Reply to comment 77: The reviewer's comment was accepted and the manuscript was revised accordingly (Line 226).

Comment 78: L226 Delete 'down'

Reply to comment 78: The reviewer's comment was accepted and the manuscript was revised accordingly (Line 232).

Comment 79: L228 Delete 'as well'

Reply to comment 79: The reviewer's comment was accepted and the manuscript was revised accordingly (Line 234).

Comment 80: L237-238 Delete 'different scales and magnitudes of the'

Reply to comment 80: The reviewer's comment was accepted and the manuscript was revised accordingly (Line 236).

Comment 81: L238-239 Change the first part of the sentence so is reads, The nitrate concentration time . . . Reply to comment 81: The reviewer's comment was accepted and the manuscript was revised accordingly (Line 237).

Comment 82: L240 After 'surface' add 'in 2011 and 2012'

Reply to comment 82: The reviewer's comment was accepted and the manuscript was revised accordingly (Line 239).
Comment 83: L241 Replace âĹij with 'about' Appears elsewhere in the manuscript.

Reply to comment 83: The reviewer's comment was accepted and the manuscript was revised accordingly (Lines 240, 249, 251, 265).

Comment 84: L246 'with higher' and delete 'times'

Reply to comment 84: The reviewer's comment was accepted and the manuscript was revised accordingly (Lines 244-245).

Comment 85: L247 . . . then followed by a reduction . . .

Reply to comment 85: The reviewer's comment was accepted and the manuscript was revised accordingly (Line 246).

Comment 86: L248 . . . scale in Fig. 3a, . . ..

Reply to comment 86: The reviewer's comment was accepted and the manuscript was revised accordingly (Line 247).

Comment 87: L249-250 . . .fluctuated neat 600 mg/L. Then concentration increased to about 32000 mg/L . . ..

Reply to comment 87: The reviewer's comment was accepted and the manuscript was revised accordingly (Line 248 - 249). We believe that the reviewer meant to the word 'near' rather to 'neat'.

Comment 88: L251 Replace 'tremendous' with 'relatively large'

Reply to comment 88: The reviewer's comment was accepted and the manuscript was revised accordingly (Line 250).

Comment 89: L252 Delete the comma and change âĹij to 'about'

Reply to comment 89: The reviewer's comment was accepted and the manuscript was revised accordingly (Line 251).

Comment 90: L254 Delete 'the lower value of'

Reply to comment 90: The reviewer's comment was accepted and the manuscript was revised accordingly (Line 253).

Comment 91: L258 . . .. migration deeper into the vadose . . ..

Reply to comment 91: The reviewer's comment was accepted and the manuscript was revised accordingly (Line 269).

Comment 92: L259 Replace 'could' with 'can'

Reply to comment 92: The reviewer's comment was accepted and the manuscript was revised accordingly (Line 269).

Comment 93: L259 Delete (Fig. 3)

Reply to comment 93: The reviewer's comment was accepted and the manuscript was revised accordingly (Line 270).

Comment 94: L260 . . . of 2.7, 4.2, 9.5, and 15.6 m . . ...

Reply to comment 94: The reviewer's comment was accepted and the manuscript was revised accordingly (Line 271).

Comment 95: L260 Replace 'escalation' with 'increase'

Reply to comment 95: The reviewer's comment was accepted and the manuscript was revised accordingly (Line 271).

Comment 96: L261 Change the comma to a semi-colon, add a comma after 'whereas' and delete m

Reply to comment 96: The reviewer's comment was accepted and the manuscript was revised accordingly (Line 272).

Comment 97: L262 Replace 'significant' with 'major'

Reply to comment 97: The reviewer's comment was accepted and the manuscript was revised accordingly (Line 272).

Comment 98: L262 Delete 'during this period'

Reply to comment 98: The reviewer's comment was accepted and the manuscript was revised accordingly (Line 273).

Comment 99: L263 Add a comma after 'period' and delete the comma after '2013'

Reply to comment 99: The reviewer's comment was accepted and the manuscript was revised accordingly (Lines 273 - 274).

Comment 100: L265 Replace 'on' with 'in'

Reply to comment 100: The reviewer's comment was accepted and the manuscript was revised accordingly (Line 275).

Comment 101: L267 Delete 'domain is'

Reply to comment 101: The reviewer's comment was accepted and the manuscript was revised accordingly (Line 277). Comment 102: L268 Delete 'm down'

Reply to comment 102: The reviewer's comment was accepted and the manuscript was revised accordingly (Line 278).

Comment 103: L269 Replace 'consists' with 'consisted'

Reply to comment 103: The reviewer's comment was accepted and the manuscript was revised accordingly (Line 279).

Comment 104: L279-280 Delete this sentence Nitrogen . . .. (Fig. 4).

Reply to comment 104: We wanted to make it clear that the water samples extracted from the sampling ports were analyzed for nitrate isotopic signature. Moreover, the other reviewer suggested adding a citation concerning this analysis.

Comment 105: L284 Replace 'like' with 'as'

Reply to comment 105: We omitted this part from the manuscript.

Comment 106: L286 Replace the first half of the sentence. At the study site, measurements showed leaching and migration of a . . .

Reply to comment 106: We omitted this part from the manuscript.

Comment 107: L296 Add '(Eq. 2) after 'calculations'

Reply to comment 107: The reviewer's comment was accepted and the manuscript was revised accordingly (Line 315).

Comment 108: L296 Replace 'a drastic' with 'an'

Reply to comment 108: The reviewer's comment was accepted and the manuscript was revised accordingly (Line 315).

Comment 109: L297 . . . increase from 2009 to 2010, at the same time as NO3 concentration increased in the upper . . .

Reply to comment 109: The reviewer's comment was accepted and the manuscript was revised accordingly (Line 316).

Comment 110: L299 . . . cultivation of the pea crop and excessive . . .

Reply to comment 110: The reviewer's comment was accepted and the manuscript was revised accordingly (Line 318).

Comment 111: L314 Delete 'Close examination of the' The results . . . ...

Reply to comment 111: The reviewer's comment was accepted and the manuscript was revised accordingly (Line 331).

Comment 112: L319 After 'model' add '(Eq. 1)'

Reply to comment 112: The reviewer's comment was accepted and the manuscript

was revised accordingly (Line 335).

Comment 113: L321 Replace 'found in' with 'applied to' Reply to comment 113: We omitted this part from the manuscript.

Comment 114: L324-326 Replace âĹij with 'about'

Reply to comment 114: See reply to comment 83.

Comment 115: L327 Replace 'over' with 'for'

Reply to comment 115: The reviewer's comment was accepted and the manuscript was revised accordingly (Line 343).

Comment 116: L336-337 Delete the last part of the sentence after 'understood'

Reply to comment 116: The practical implementation section was revised and this line was moved to the beginning of the first paragraph of this section.

Comment 117: L337 Replace 'Today's' with 'The'

Reply to comment 117: We omitted this part from the manuscript. See reply to comment 116.

Comment 118: L338 Replace 'might' with 'may'

Reply to comment 118: We omitted this part from the manuscript. See reply to comment 116.

Comment 119: L345 Replace 'which' with 'that'

Reply to comment 119: We omitted this part from the manuscript. See reply to comment 116.

Comment 120: L346 Replace 'by the distribution' with 'with'

Reply to comment 120: We omitted this part from the manuscript. See reply to comment 116.

Comment 121: L347 Replace 'impact' with 'effects'

Reply to comment 121: We omitted this part from the manuscript. See reply to comment 116.

Comment 122: L540 Delete the first footnote. It is not needed as the heading in the table already indicates this.

Reply to comment 122: The reviewer's comment was accepted and the manuscript was revised accordingly (Line 630).

Comment 123: L546-549 Show Fig. 1c as a separate diagram. The diagram shows an observational well. There is no mention of this well in the methods or elsewhere in the paper. Please remove from the diagram. Show a distance scale in the diagram to indicate that the water table is about 18 m below the soil surface.

Reply to comment 123: The reviewer's comment was partly accepted. We removed the observation well from the diagram and added the distance scale. However, we insisted on keeping the vadose zone monitoring illustration underneath the pictures of the study site, since we think it describes well the system installation and implementation within the field.

Comment 124: L551-552 Figure 2. Water-content (o) at different depths in the vadose zone and daily rainfall for six consecutive years.

Reply to comment 124: The reviewer's comment was accepted and the manuscript was revised accordingly (Line 641 - 642).

Comment 125: L554-556 Figure 3. Time series of observed (NO3) concentrations in the vadose zone and daily rainfall for six consecutive years.

Reply to comment 125: The reviewer's comment was accepted and the manuscript was revised accordingly (Line 644 - 645).

Comment 127: L561-562 . . .. entire vadose zone per year.

Reply to comment 127: The reviewer's comment was accepted and the manuscript was revised accordingly (Line 650).

Comment 128: L564 Delete '(red circle)' and '(dashed blue line)' Figs. 2 and 3 The text in the these two figures seem to be stretched. Please re-size the figures.

Reply to comment 128: We deleted the legends in the figure so the captions did not change. The figures were re-size.

---

## Editor Decision (ED1)

**Abstract**

Nitrate is considered the most common non-point pollutant in groundwater. It is often

attributed to agricultural management, when excess application of nitrogen fertilizer

leaches below the root zone and is eventually transported as nitrate through the

unsaturated zone to the water table. A lag time of years to decades between processes

occurring in the root zone and their final imprint on groundwater quality prevents

proper decision-making on land use and groundwater-resource management. This

study implemented the vadose monitoring system (VMS) under a commercial cropfield. Data obtained by the VMS for of 6 years allowed, for the first time known to us,

a unique detailed tracking of water percolation and nitrate migration from the surface

through the entire vadose zone to the water table at 18.5 m depth. A nitrate

concentration time series, which varied with time and depth, revealed—in real time—

a major pulse of nitrate mass propagating down through the vadose zone from the root

zone toward the water table. Analysis of stable nitrate isotopes indicated that manure

is the prevalent source of nitrate in the deep vadose zone and nitrogen transformation

processes have little effect on nitrate isotopic signature. The total nitrogen mass

calculations emphasized the nitrate mass migration towards the water table.

Furthermore, the simulated pore-water velocity through analytical solution of the

convection–dispersion equation shows that nitrate migration time from land surface to

groundwater is relatively rapid, approximately 5.9 years. Ultimately, agriculture land

uses, which are constrained to high nitrogen application rates and coarse soil texture,

are prone to induce substantial nitrate leaching.

Page: 2

Number: 1 Author:     Subject: Inserted Text     Date: 6/28/16, 10:47:41 AM

groundwater recharge behavior and tendency in the deep vadose zone of two agricultural settings, a grapefruit orchard and a crop field (Turkeltaub et al., 2014). Unsaturated flow models were calibrated to the water content observation and were used for groundwater recharge fluxes simulations.

The objective of the present study was to demonstrates the water flow and nitrate transport through the deep vadose zone underlie the crop field, with respect to rain patterns as well as the agricultural and fertilization setup. Continuous data on variations in the sediment water content and nitrate concentrations were collected from the entire vadose zone for over 6 years. The nitrate concentration time series, which included variation of nitrate in time and at multiple depths, revealed, in real time, a major pulse of nitrate mass propagating down through the vadose zone toward the water table. These results indicate that nitrate fluxes in the unsaturated zone underlie agriculture land-uses were associated with high nitrogen application rates and coarse texture soils. Furthermore, pollution events originated from agriculture land-uses can be monitored in their early stages, long before pollution accumulates in the aquifer water.

**2 Methods**

**2.1 Study area**

A commercial crop field site was selected as a representative prevalent agriculture setting in the southern part of the coastal plain of Israel (34°41'13" E; 31°49'42" N) and is part of an array of VMSs that were installed under different representative land-uses situated above the southern part of the phreatic costal aquifer

**Page: 6**

(Dahan et al., 2014, Baram et al., 2013, 2014, Turkeltaub et al., 2014,2015a, 2015b).

The study was conducted between 09/2009 and 04/2015 [1]Mediterranean climate

prevails in this area, with hot, dry summers (May–September) and rainy winters

(October–April), with an average annual rainfall of 512 mm and average temperatures

of 31.2 °C (August) and 17.8 °C (January) in the hottest and coldest months,

respectively (Israeli Meteorological Service, 2015). Reference evapotranspiration

rates calculated according to the Penman–Monteith method (suggested by the Food

and Agriculture Organization) range from 1.5 mm $day^{-1}$ (January) to 5.7 mm $day^{-1}$

(July) (Israeli Meteorological Service, 2015).

The crop field cultivation history includes alternation between rainfed

agriculture, as [2]heat [3]nd irrigated agriculture as [4]atermelon for seeds [5]nd cotton as

summer crop ([6]ersonal communication). From 2005 to 2013, the crop field site was

cultivated with rainfed winter crops—spring wheat (*Triticum aestivum* L.) and pea

(*Pisum sativum* L.) (Fig. 1). Then for 1 year (2013/2014), the field was uncultivated.

The crops were sown at the beginning of the wet season (November) and grew into

the spring (April). After harvest, disk plow and roller practices were implemented.

Since 2005, the main fertilization application to the field was dairy-farm slurry

manure, which was distributed over the 10 ha field for 60 days during May and June

(Fig. 1). The total nitrogen concentration in the dairy slurry is 900 mg $L^{-1}$ (Water

Authority, 2012). In September 2014, jojoba (*Simmondsia chinensis*) shrubs were

planted and irrigation systems were installed.

**2.2 Monitoring**

**Page: 7**

.

with

,

ith

,

State who the personal communication is from and the date. Use proper format.

year[-1] (Herridge et al., 2008), which is about 43% of the nitrogen applied by the dairy slurry. Thus, application of dairy farm slurry combined with a legume crop (pea) seemed to have enriched the top soil with excess nitrogen, as compared to cultivation of cereal-type crops (Fig. 3a).

Progression of the nitrate migration deeper into the vadose zone can be divided into two periods. In the first period, October 2010 to January 2013, at depths of 2.7, 4.2, 9.5 and 15.6 m (Fig. 3b,c,e,g), the increase in nitrate concentration was moderate and continuous; whereas, at depths of 6.3 and 18 m, there was no major change in nitrate concentrations (Fig. 3b-d). In the second period, starting from July 2013 following the rainy winter of 2012/13, substantial nitrate breakthroughs were noticeable throughout most of the vadose zone cross section (marked with arrows in Fig. 3). This rapid nitrate progression to the deeper parts of the vadose zone could be related to the soil's physical characteristics. In the top 3 m, the soil comprised of fine-textured layers (sandy-loam and loamy sand), and from 3 to 18.5 m (water table), the soil consisted of a coarser sand-textured layer (Turkeltaub et al., 2014). Thus, as a consequence of substantial water percolation, which induced intensive water flux across the coarse-textured soil, nitrate transport could be detected at deeper depths of the vadose zone.

Here, as well in previous studies in literature, nitrate fluxes in the unsaturated zone underlie agriculture land-uses were associated with nitrogen application rates and soil physical properties (Green et al., 2008; Botros et al., 2012; Turkeltaub et al., 2015b). Therefore, to attenuate nitrate leaching to aquifers, search should be dedicated to locate the 'hot spots' where these conditions prevailed (Liao et al., 2012).

**3.2 Nitrate sources**

**Page: 13**

Number: 1 Author: Subject: Cross-Out Date: 6/28/16, 11:25:21 AM

Number: 2 Author: Subject: Inserted Text Date: 6/28/16, 11:25:32 AM
site characterization efforts

Number: 3 Author: Subject: Cross-Out Date: 6/28/16, 11:25:37 AM

Number: 4 Author: Subject: Cross-Out Date: 6/28/16, 11:26:32 AM

Number: 5 Author: Subject: Inserted Text Date: 6/28/16, 11:25:42 AM
ing

Number: 6 Author: Subject: Inserted Text Date: 6/28/16, 11:26:58 AM
conditions favor higher rates of transport

[revised manuscript text omitted]

Number: 1 Author:     Subject: Highlight      Date: 6/28/16, 12:59:42 PM
This seems misleading because it implies that in this study, data from an array of VMSs was used; but my understanding is that the data in this paper are only from ONE VMS. I think you need to replace "are combined" to "will be combined in the future".

---

## Author Response (AR2)

Dear Graham Fogg

We received the review of our manuscript entitled "**Real-time monitoring of nitrate transport in deep vadose zone under a crop field—implications for groundwater protection**" by Turkeltaub et al (hess-2016-63). We want to thank you for your work in improving the English in our paper and all your comments were addressed.

Sincerely,

Tuvia Turkeltaub